# Acupuncture on ST36, CV4 and KI1 Suppresses the Progression of Methionine- and Choline-Deficient Diet-Induced Nonalcoholic Fatty Liver Disease in Mice

**DOI:** 10.3390/metabo9120299

**Published:** 2019-12-09

**Authors:** Xiangjin Meng, Xin Guo, Jing Zhang, Junji Moriya, Junji Kobayashi, Reimon Yamaguchi, Sohsuke Yamada

**Affiliations:** 1Department of Pathology and Laboratory Medicine, Kanazawa Medical University, 1-1 Daigaku, Uchinada, Kahoku, Ishikawa 920-0293, Japan; meng6950@kanazawa-med.ac.jp (X.M.); jing2018@kanazawa-med.ac.jp (J.Z.); sohsuke@kanazawa-med.ac.jp (S.Y.); 2Department of General Internal Medicine, Kanazawa Medical University, Ishikawa 920-0293, Japan; moriya@kanazawa-med.ac.jp (J.M.); mary@kanazawa-med.ac.jp (J.K.); 3Department of Pathology, Kanazawa Medical University Hospital, Ishikawa 920-0293, Japan; 4Department of Dermatology, Kanazawa Medical University, Ishikawa 920-0293, Japan; raymon-y@kanazawa-med.ac.jp

**Keywords:** nonalcoholic fatty liver disease, acupuncture, imflammation, lipid metabolism, oxidative stress

## Abstract

Nonalcoholic fatty liver disease (NAFLD) is one of the most common chronic liver diseases worldwide, and its treatment remain a constant challenge. A number of clinical trials have shown that acupuncture treatment has beneficial effects for patients with NAFLD, but the molecular mechanisms underlying its action are still largely unknown. In this study, we established a mouse model of NAFLD by administering a methionine- and choline-deficient (MCD) diet and selected three acupoints (ST36, CV4, and KI1) or nonacupoints (sham) for needling. We then investigated the effects of acupuncture treatment on the progression of NAFLD and the underlying mechanisms. After two weeks of acupuncture treatment, the liver in the needling-nonapcupoint group (NG) mice appeared pale and yellowish in color, while that in the needling-acupoint group (AG) showed a bright red color. Histologically, fewer lipid droplets and inflammatory foci were observed in the AG liver than in the NG liver. Furthermore, the expression of proinflammatory signaling factors was significantly downregulated in the AG liver. A lipid analysis showed that the levels of triglyceride (TG) and free fatty acid (FFA) were lower in the AG liver than in the NG liver, with an altered expression of lipid metabolism-related factors as well. Moreover, the numbers of 8-hydroxy-2′-deoxyguanosine (8-OHdG)-positive hepatocytes and levels of hepatic thiobarbituric acid reactive substances (TBARS) were significantly lower in AG mice than in NG mice. In line with these results, a higher expressions of antioxidant factors was found in the AG liver than in the NG liver. Our results indicate that acupuncture repressed the progression of NAFLD by inhibiting inflammatory reactions, reducing oxidative stress, and promoting lipid metabolism of hepatocytes, suggesting that this approach might be an important complementary treatment for NAFLD.

## 1. Introduction

Nonalcoholic fatty liver disease (NAFLD) is one of the most common clinicopathological conditions in chronic liver disease and is characterized by an obvious increase in fat deposition in the hepatocytes of the liver parenchyma [1]. Epidemiologically, NAFLD affects approximately 40% of the population worldwide, and the prevalence is increasing annually all over the world [2]. This disease has been considered to carry a high risk of developing into liver cirrhosis and hepatocellular carcinoma.

However, there are still no effective drugs specifically developed for the treatment of NAFLD, especially in patients with non-alcoholic steatohepatitis (NASH), a severe stage of NAFLD. Although several agents have shown some benefits for patients with NAFLD, even the most promising of such pharmacological agents are associated with significant adverse effects, and none have been approved by the Food and Drug Administration (FDA) for NAFLD therapy [3]. Balancing the benefits and risks in drugs for long-term treatment remains a constant challenge that has hampered the development of therapy strategies for NAFLD [4].

Ectopic fat accumulation in the hepatocytes, a hallmark of NAFLD, is thought to be caused by a complex and multiple mechanism that is still not completely understood but involves several interdependent molecular processes, such as inflammation, lipid metabolism, and oxidative stress [5]. An abnormal lipid metabolism, such as an increased uptake of lipids and the upregulation of de novo lipogenesis in the liver, is the initial trigger of NAFLD. Lipid overload promotes reactive oxygen species (ROS) generation and peroxidation itself, which causes the release of pro-inflammatory cytokines and inflammatory cellular infiltration [6,7]. The activation of the nuclear factor-κB (NF-κB) inflammatory pathway regulating downstream target genes plays very important roles in the progression of NAFLD. These inflammatory cytokines can recruit and activate Kupffer cells/macrophages and further aggravate liver injury and steatohepatitis formation [8].

Oxidative stress caused by the imbalance between oxidants and antioxidants seems to be one of the most important mechanisms leading to NAFLD liver injury, which plays a fundamental role in the progression from simple steatosis to NASH. The enhanced production of ROS can reportedly lead to necroinflammation and fibrosis through lipid peroxidation induced by astrocyte activation [9].

Acupuncture is a treatment method found in traditional Chinese medicine (TCM). Due to its advantages of low cost, few side effects, and simple operation, the role of acupuncture in disease prevention and treatment has recently attracted attention, and a large body of evidence has shown that acupuncture can induce pathophysiological consequences and alleviate the symptoms of diseases in multiple organs [10,11,12]. A number of clinical trials have also indicated that acupuncture treatment can improve metabolism conditions and exert beneficial effects on patients with NAFLD [13,14,15], but the mechanisms underlying its action remain unclear.

According to the theory of TCM, fatty liver formation primarily involves metabolic disorders caused by qi and blood stasis, and acupuncture at certain points of related meridians can improve the metabolism by keeping the body in a balanced state between “Yin” and “Yang” [16]. However, from a modern medicine viewpoint, how acupuncture exerts its therapeutic role and what kind of physiological consequences acupuncture causes at different acupoints of meridians are still unclear.

In the present study, we established a mouse model of NAFLD by administering a methionine- and choline-deficient diet (MCD), a classic diet inducing NAFLD [17]. We selected three acupoints of related meridians regulating the metabolism to needle the mice with NAFLD and observed the roles of acupuncture in the progression of NAFLD. Furthermore, we also investigated the pathogenic mechanisms leading to these significant effects through laboratory and molecular biology experiments. Our results suggest that acupuncture might be a useful treatment for NAFLD and provide solid evidence supporting the incorporation of acupuncture into therapy for metabolic syndrome from a modern medicine viewpoint.

## 2. Results

### 2.1. Lipid Accumulation was Significantly Reduced in the Livers of AG Mice with NAFLD Induced by an MCD Diet

After two weeks of acupuncture, the livers from the needling-nonapcupoint group (NG) mice were variably pale and yellowish in color. In contrast, the livers from the needling-acupoint group (AG) showed a bright red color (Figure 1A). The mouse body weight and liver weight were significantly lower in AG mice than in those of NG mice (*p* < 0.0001, *n* = 17) (Appendix A). However, the mouse liver/BW ratios were no significant difference between the two groups (*p* < 0.0001, *n* = 17) (Figure 1B).

Hematoxylin and eosin (H&E) staining showed that the AG liver tended to contain fewer lipid droplets and inflammatory foci than those of NG mice, and there was also a significant histological difference between the two livers. Ballooning of hepatocytes, inflammation, and fibrosis were noted in the livers of NG mice. Furthermore, the NASH score in the AG livers was significantly lower than that in the NG livers after acupuncture (*n* = 17) (Figure 2A and Table 1). Oil red-O staining revealed few lipid droplets in the AG mice, while the numbers of lipid-positive hepatocytes and droplets in each hepatocyte were increased in NG mice (*n* = 17) (Figure 2B). These morphology and histology findings indicate that acupuncture can treat the liver damage caused by a high-fat diet.

### 2.2. Acupuncture Treatment Inhibits the Inflammation Reaction during the Progression of MCD Diet–Induced NAFLD

IHC for Mac-2 revealed that the AG livers contained a significantly smaller number of infiltrating macrophages (Kupffer cells) than the NG livers (AG 6.7 ± 2.5 vs. NG 12.3 ± 4.7; *p* < 0.001, *n* = 17) (Figure 3A). In addition, fibrogenesis and stellate cell activation, as determined by IHC for α-smooth muscle actin (α-SMA), were not apparent in the AG liver, and the specific and linear expression of α-SMA along with the activation of hepatic stellate cells in AG mice was much lower than that in NG mice (AG 18.7 ± 6.1.0 vs. NG 39.4 ± 7.5; *p* < 0.0001, *n* = 17) (Figure 3B).

A Western blot analysis showed that the expression of proinflammatory signaling factors and inflammatory cytokines or transcriptional factor and their receptors, such as interleukin 1β (IL-1β), tumor necrosis factor-α (TNFα) and the key regulators of inflammation, NF-κB and phospho-nuclear factor-ΚB (p-NF-κB), was significantly lower in the AG livers than in the NG livers. Therefore, acupuncture can promote liver metabolism by inhibiting inflammatory reaction (*n* = 7) (Figure 3C).

### 2.3. Acupuncture Treatment Changed the Lipid Profiles and Regulated Lipid Metabolism in the Liver with NAFLD Induced by an MCD Diet

The hepatic triglyceride (TG) and free fatty acid (FFA) levels in liver were significantly lower in AG mice than in NG mice (TG: AG 6.7 ± 3.5 mg/g vs. NG 11.0 ± 5.2 mg/g; *p* < 0.05, *n* = 17; FFA: AG 0.15 ± 0.15 mEq/g vs. NG 0.4 ± 0.2 mEq/g; *p* < 0.01, *n* = 17). However, the liver T-cho levels were not significantly different between AG and NG mice (Figure 4A).

Real Time Reverse Transcription Polymerase Chain Reaction (RT-PCR) showed that no marked differences between the two groups were observed in the expression of some receptors related to the lipid uptake in the liver, including low-density lipoprotein receptor (LDLR), scavenger receptor class A (SR-A) and scavenger receptor class B type 1 (SR-B1) (Figure 4B). However, the expression of transcription factor sterol regulatory element binding protein 1 (SREBP1), an important transcriptional protein that regulates lipid synthesis with a well-studied function in lipid metabolism [18], was significantly lower in the AG livers than in the NG livers (*p* < 0.05, *n* = 7), while that of SREBP2, which primarily regulates cholesterol biosynthesis, showed no significant difference between the two groups (Figure 4B). The expression of SREBP1 target genes, such as fatty acid synthase (FAS) and stearoyl-CoA 9-desaturase 1 (SCD1), was also significantly lower in the AG livers than in the NG livers (*p* < 0.05, *n* = 7) (Appendix A). Peroxisome proliferator-activated receptors (PPARs) are primary modulators in the metabolism of fatty acids in the liver [19] and include PPARα, PPARβ/δ, and PPARγ. The PPARγ RNA expression was significantly lower in AG livers than NG livers (*p* < 0.05, *n* = 7), as well as the the expression of PPARγ target such as adiponectin receptor 2 (AdipoR2). The expression was significantly lower in AG livers than NG livers (*p* < 0.05, *n* = 7) (Appendix A), but the PPARα and PPARβ/δ RNA expression did not differ significantly between the groups (Figure 4B). Moreover, the expression of genes involved in hepatic lipid secretion apolipoprotein B (ApoB), apolipoprotein E (ApoE), and microsomal triglyceride transfer protein (MTTP) were significantly higher in the AG livers than in the NG livers (*p* < 0.05, *n* = 7) (Appendix A).

### 2.4. Acupuncture Treatment Improved Oxidative Stress Induced by Lipid Accumulation of the NAFLD Liver in Mice

We used IHC to determine the expression of 8-hydroxy-2′-deoxyguanosine (8-OHdG) as a marker for oxidative stress. There were significantly fewer cells positive for 8-OHdG in AG mice than in NG mice after 2 weeks of acupuncture (AG: 104.5 ± 26.4 vs. NG: 221.8 ± 63.9; *p* < 0.001, *n* = 17) (Figure 5A).

We next measured the liver levels of another oxidative stress marker, thiobarbituric acid reactive substances (TBARS). The TBARS levels were significantly lower in AG mice than in NG mice after 2 weeks of acupuncture (AG 4.5 ± 2.5 nmol malondialdehyde ([MDA]) protein vs. NG 6.3 ± 3.3 nmol MDA protein, *p* < 0.05, *n* = 17) (Figure 5B).

Real time PCR showed that the hepatic expression of several antioxidant enzymes, including glutathione peroxidase 1, 2, and 3 (GPx1, 2, and 3), glutathionylspermidine synthetase/amidase (Gss), and catalase and the transcription factor nuclear factor erythroid 2-related factor 2 (Nrf2), were significant higher in AG mice than in NG mice (*p* < 0.05, *n* = 7) (Figure 5C). Therefore, acupuncture may reduce oxidative stress by upregulating the antioxidant expression.

## 3. Discussion

Acupuncture is a vital component of TCM and has a history of more than 2500 years. Given its safety and few side effects, this major TCM has been widely used to treat various diseases and symptoms, especially chronic metabolic illness [20]. Many modern studies have proven the efficacy of acupuncture against a wide range of diseases [21,22,23]. However, due to lack of understanding of the mechanisms underlying its action, acupuncture is the subject of severe controversy, with its effects proposed to be placebo effects [24]. To clarify these points and the efficacy of acupuncture on the systemic metabolism, it is necessary to determine the molecular mechanism underlying acupuncture’s effects on specific tissues and cell metabolism.

Since it is illegal and unethical to carry out laboratory experiments on the human body without need, animal studies of acupuncture are of great value and show obvious advantages regarding research on the metabolic signal pathways compared to clinical studies [25]. Therefore, in this study, we used a mouse model of MCD + HF diet-induced NAFLD to investigate the mechanisms by which acupuncture treatment improves the conditions of this chronic disease.

According to TCM, the three acupoints of Zusanli (ST36), Yongquan (KI1) and Guanyuan (CV4) were selected for needling model mice in order to harmonize the Yin and Yang and dredge the channel of Qi and Blood. Our pathological and experimental results showed that acupuncture treatment significantly attenuated the progression of NAFLD by inhibiting inflammatory reactions, reducing oxidative stress and promoting lipid metabolism of hepatocytes. These results provide solid evidence from a modern medicine perspective supporting the notion that acupuncturing these three acupoints may be beneficial for patients with NAFLD.

Inflammation plays very important roles in the process of NAFLD progression. Injured liver cells can release damage-associated molecular patterns (DAMPs) to promote the activation of the NF-κB pathway, thus inducing the production of pro-inflammatory cytokines like TNF-α and ILs, which is a key step in the progression from simple steatosis to NASH [26,27]. Many studies have confirmed that acupuncture can downregulate the NF-κB expression [28,29]. Indeed, in the present study, hepatic injury induced by the absence of methionine and choline in the diet clearly upregulated the expression and enhanced the activation of NF-κB, thus increasing the production of downstream factors TNF-α and IL-1β in the livers of NG mice, while acupuncture treatment significantly inhibited the NF-κB inflammatory signals in the AG mice. These cytokines can contribute to the recruitment and activation of macrophages/Kupffer cells (resident hepatic macrophages) to mediate inflammation [30], which is critical in NASH. A significant increase in macrophages has been shown to occur in the liver tissue of patients with NASH compared to those with simple steatosis [31]. In the present study, with the reduction in the TNF-α and IL-1β expression, much fewer Mac-2- and α-SMA-positive cells were observed in the livers of AG mice with simple steatosis than in NG mice showing progression to NASH. These results indicate that acupuncture treatment can improve the pathological progression of NAFLD by inactivating the inflammatory signaling pathways.

The regulatory role of acupuncture in inflammation has also been reported in other studies, but the effects are not the same when needling different acupoints. For example, acupuncture of the Sanyinjiao (SP6) acupoint can increase IL-10 levels [32], while needling Fenglong (ST40) and Neiguan (PC6) acupoints reduced the IL-17 expression [33]. These data imply that although acupuncture treatment at several acupoints can suppresses the inflammation response, different acupoints can achieve the same effect by regulating specific cytokines.

The accumulation of excess lipid in hepatocytes causes organelle failure, such as mitochondrial dysfunction and endoplasmic reticulum stress, and leads to liver injury in patients with NAFLD [34,35]. However, in some NAFLD patients, the accumulation of lipids is not toxic to liver cells, a paradoxical effect that is believed to be related to the type of lipid itself. For example, TG reportedly does not seem toxic, but FFA and cholesterol—including its metabolites—are highly toxic to cells [36,37]. In our MCD-induced NAFLD mouse model, the hepatic total cholesterol levels were not markedly different between NG and AG mice, but the TG and especially the FFA levels were markedly reduced in the livers of AG mice compared to the livers of NG mice. Therefore, a significant decrease in the FFA level in the liver may play a more important role in acupuncture treatment for NAFLD than a reduction in TG. The FFA pool in the circulation is the major source of FFA in the liver [38], but no remarkable difference in the hepatic expression of receptors related to the lipid uptake were noted between NG and AG mice. However, we found that some regulators expressions regarding de novo lipogenesis and lipid storage in the liver were significantly decreased and genes involved in hepatic lipid secretion increased in AG mice compared with NG mice, suggesting that the reduction in hepatic lipid deposition after acupuncture treatment is induced by reducing the synthesis and promoting the metabolism of lipids rather than by inhibiting their uptake in the liver altogether.

Mitochondrial dysfunction and endoplasmic reticulum stress caused by the accumulation of lipids, especially FFA, can also result in the increased production of ROS and lead to oxidative stress that promotes inflammatory reactions, the activation of stellate cells and fibrosis in the liver, which have been recognized as important events in the development of NAFLD [39,40,41]. In the present study, the number of 8-OHdG-positive cells and the level of MDA in the liver were significantly reduced in AG mice compared to NG mice, showing that acupuncture treatment can also improve the oxidative stress status during NAFLD progression. It seems easy to understand that the lower level of hepatic FFA inhibited organelle failure and repressed the increased production of ROS in AG mice. These results should be rational, however, the consideration is probably oversimplified, as we also found expressions of some protective antioxidants to significantly increase in the livers of AG mice (Figure 5C), which may be the real reason for the inhibition of oxidative stress by the accumulation of lipids. Another interesting finding of the present study was that acupuncture treatment had significant regulatory effects on inflammatory reaction, lipid metabolism and redox homeostasis, which was found to be closely associated with changes in the expression of transcription factors related to these signaling pathways, like NF-κB, PPARs, and Nrf2 (Figure 3C, Figure 4B, and Figure 5C respectively). These results remind us that acupuncture may regulate cell metabolism at the expression level. To confirm this suspicion, more molecular mechanism experiments will be needed in the future.

In summary, from this study, we obtained some novel findings: (1) acupuncture on the three acupoints, ST36, CV4 and KI1, can improve pathological process of NAFLD; (2) acupuncture treatment can inhibit inflammatory reactions, reduce oxidative stress and promote lipid metabolism; (3) acupuncture on different acupoints can inhibit inflammatory reactions by regulating specific cytokines; (4) acupuncture treatment have regulatory effects on the expression of transcription factors. Although some exciting results were found in the present study, there are still some limitations to be noted. First, an MCD + HF diet-induced NAFLD model was used to investigate the roles of acupuncture in inhibiting the progression of NAFLD in this study, but the molecular mechanism underlying special diet-induced NAFLD does not totally reflect the pathogenesis of this disease in human, even though some pathological manifestations are consistent between the two entities. Second, although acupuncture treatment has few side effects, since this was the first instance of observing the effects of needling the three acupoints ST36, KI1, and CV4 in mice with NAFLD, and since this treatment was administered daily for only two weeks, we cannot confirm that no side effects would be noted with this approach over a long period of time. Finally, after two weeks of acupuncture treatment, we also found metabolic changes in other tissues and organs aside from the liver, largely related to gastrointestinal absorption, lipid storage and energy metabolism (data not shown), which have a major influence on metabolic syndrome, such as insulin resistance, obesity and fatty liver. However, in the present study, we ignored these other influences temporarily and instead focused on the metabolic changes in the liver after acupuncture treatment. The influence of other factors should therefore be discussed in future studies.

## 4. Materials and Methods

### 4.1. Animals and Experimental Protocol

Experiments were performed using 8-week-old male C57BL/6 mice weighing approximately 20 g that were maintained in a temperature- and light-controlled facility with free access to water. Mice were fed an MCD + HF diet (60% fat; KBT Oriental Corporation, Saga, Japan) for 3 weeks and then given an HF diet for two weeks to maintain their hyperlipidemia. As described previously [17], mice were anesthetized with an injection of ketamine-medetamidine and euthanized by exsanguination. The liver was excised and cut into small pieces, frozen, and fixed in 10% neutral-buffered formalin for the experiments described below.

### 4.2. Acupuncture Manipulation

The mice were randomly divided into two groups: AG and NG. They were fed an MCD + HF diet for three weeks and then given HF diet for two weeks to maintain their hyperlipidemia. For the needling treatment, three acupoints (AG) or no acupoints (NG) were needled (Figure 6A).

The ST36 acupoint is located near the knee joint of the hind limb and 1.5 mm from the distal side of the anterior tibial tubercle. The KI1 acupoint is located in the middle of the hind paw. The pubic symphysis was obliquely stabbed at point CV4 at the mouse abdomen median line, 10 mm below the navel [22,42,43] (Figure 6B). AG mice received acupuncture at both sides of ST36 and CV4 with 13-mm needles. Acupuncture was delivered using stainless steel needles (length: 13 mm, diameter: 0.25 mm; Hwatuo, Suzhou Medical Supplies Factory Co., Ltd., Suzhou, China). ST36 and CV4 needling was performed by straightly inserting a stainless steel needle to a depth of 3 mm. KI1 was needled obliquely toward the elbow to a depth of 2 mm. NG mice received nonacupoint needling. All points were rotated slowly at 60 rounds per minute, completed in 2 min, without retaining the needle.

### 4.3. Ethics

The Ethics Committee of Animal Care and Experimentation, Kanazawa Medical University, Japan, approved the protocols. The project code of the approval, 2019-21, was identified on 3 July 2019. Experiments were performed according to the Institutional Guidelines for Animal Experiments and the Law (no. 105) and Notification (no. 6) of the Japanese government. The number of animals used and their suffering were minimized.

### 4.4. Histopathology

After fixation in 10% neutral-buffered formalin for 24 h, paraffin-embedded liver specimens were systematically cut into sequential 4-μm-thick sections. For histological analyses of the liver, images of hematoxylin and eosin (H&E), oil red-O, and immunohistochemistry.

IHC sections were captured and quantified using the NanoZoomer Digital Pathology Virtual Slide Viewer software program (Hamamatsu Photonics Corp, Hamamatsu, Japan). To evaluate the degree of lipid accumulation (steatosis score and lipid accumulation score for the liver), we performed oil red-O staining using frozen liver sections and categorized the tissues into 4 grades, as follows: no lipid droplets (score = 0); lipid droplets in <33% of hepatocytes (score = 1); lipid droplets 33%–66% of hepatocytes (score = 2) and lipid droplets in >66% of hepatocytes (score = 3). In addition, the degree of liver cell ballooning injury (ballooning score) was classified into three grades as follows: none (score = 0); few balloon cells (score = 1) or many balloon cells/prominent ballooning (score = 2).

### 4.5. IHC

To evaluate the severity of NAFLD, we determined the intensity of inflammation (inflammatory score) using an anti-mouse Mac-2 monoclonal antibody (1:1000; Cedarlane Laboratories Ltd., Burlington, Ontario, Canada). As described previously [44], we counted the number of positive macrophages in 10 randomly selected fields per liver section (original magnification: ×200). The NAFLD liver tissues were then classified into four (inflammation score) grades, as follows: no inflammation (score = 0); <10 inflammatory foci, each consisting of >5 inflammatory cells (score = 1); ≥10 inflammatory foci (score = 2) or uncountable diffuse or fused inflammatory foci (score = 3). We used the HistoMouse™–Plus Kit (Invitrogen Corporation, Camarillo, CA, USA) to block endogenous IgG and then stained tissue with a monoclonal mouse anti-human α-SMA antibody (1:1000; Dako Cytomation, Carpenteria, CA, USA.). The number of activated stellate cells was then counted in 10 randomly selected fields per section (original magnification: ×200), as described previously [44]. To determine the ROS/oxidative stress or expression in hepatocytes after acupuncture, we used an 8-OHdG monoclonal antibody (1:200; Japan Institute for the Control of Aging, Fukuroi, Japan) and quantified the number of hepatocytes positive for either antibodies in 10 randomly selected fields per section (original magnification: ×200), as previously described. For IHC studies, we examined at least 1 section from each of 17 mice per experimental group.

All histological and immunohistochemical slides were evaluated by two independent observers (certified surgical pathologists in our department; X.G. and S.Y.) using a blind protocol design (observers blinded to the mice treatment data). The agreement between the observers was excellent (more than 95%) for all antibodies investigated.

### 4.6. Western Blotting

Liver protein samples were separated by electrophoresis on 10% sodium dodecyl sulfate-polyacrylamide gel electrophoresis (SDS-PAGE) gels and transferred onto Immun-Blot PVDF membranes (Bio-Rad Laboratories, K.K., Tokyo, Japan). The membranes were then incubated overnight at 4 °C with IL-1β antibody (#12242; Cell Signaling,), TNF-α antibody (#11948; Cell Signaling), NF-κB antibody (#8242; Cell Signaling), phospho-NF-κB antibody (#3033; Cell Signaling) and anti-β-actin monoclonal antibody (Wako Pure Chemical Co., Osaka, Japan) diluted in Can Get Signal solution 1 (Toyobo, Osaka, Japan), after which the membranes were incubated for 1 h at room temperature with a horseradish peroxidase-conjugated goat anti-rabbit antibody (Vector Laboratories, Burlingame, CA, USA).

### 4.7. Analyses of Lipid Contents from the Liver

To examine the hepatic lipid profiles, each snap-frozen tissue (70 mg) was homogenized and extracted with chloroform-methanol (2/1 v/v), as described previously [45]. The organic phase was dried and resolubilized in 2-propanol. The TG, FFA, and TCHO levels were then determined using commercial assay kits (Wako Pure Chemical Co.)

### 4.8. Real Time Reverse Transcription Polymerase Chain Reaction (RT-PCR)

Real time PCR was used to analyze the gene expression in the liver. Total RNA was extracted from mouse liver using the extracted by ReliaPrep™ RNA Tissue Miniprep kit (Promega, Leiden, Netherlands). The whole extraction process was performed under RNase-free conditions in order to prevent RNA degradation. Custom primers and TaqMan probe for gene amplification were purchased from Life Technologies. The mRNA expression of SREBP1, LDLR, SR-A, SREBP2, SR-B1, PPARγ, PPARα, PPARβ/δ, GPx1, GPx2, GPx3, Gss, catalase, and Nrf2 was analyzed by real time PCR (TaqMan probes Applied Biosystems, Warrington, UK). The relative expression of each gene was normalized to that of 18S ribosomal RNA using random primers.

### 4.9. The Measurement of the TBARS Levels

We measured the liver TBARS levels using a TBARS Assay Kit (Cayman Chemical Company, Ann Arbor, MI, USA). Liver tissue specimens were homogenized in 250 RIPA buffer solution. A 100-μL aliquot of the homogenate was added to a reaction mixture containing 200 μL of 8.1% (w/v) SDS, 1.5 mL of 20% (v/v) acetic acid, pH 3.5, 1.5 mL of 0.8% (w/v) thiobarbituric acid, and 700 μL of distilled water. Samples were then boiled for 1 h at 95 °C and centrifuged at 1600× *g* for 10 min. The absorbance of the supernatant was measure spect rophotometrically at a wavelength of 530–540 nm [46].

### 4.10. Statistical Analyses

The results are expressed as the means ± standard deviation (SD). Significant differences were analyzed using Student’s t-test, Welch’s t-test or a one-way analysis of variance (ANOVA), where appropriate. Values of *p* < 0.05 were considered to be statistically significant.

## Figures and Tables

**Figure 1 metabolites-09-00299-f001:**
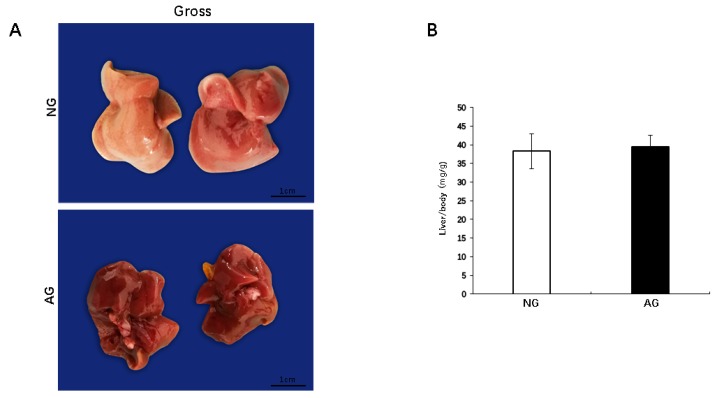
The appearance of the liver and ratio of the liver to the body weight after two weeks of acupuncture. (**A**) The liver tissues of the NG mice were obviously pale and yellowish in color, and while the tissue of the AG liver showed a bright red color. Bar = 1 cm. (**B**) There was no marked difference in the ratio of the liver to the body weight between the two groups. Values are shown as the mean ± SD, * *p* < 0.05, ** *p* < 0.001, *** *p* < 0.0001, *n* = 17. NG: needling-nonacupoint group, AG: needling-acupoint group, MCD diet: methionine- and choline-deficient diet, HF diet: high-fat diet, NAFLD: nonalcoholic fatty liver disease.

**Figure 2 metabolites-09-00299-f002:**
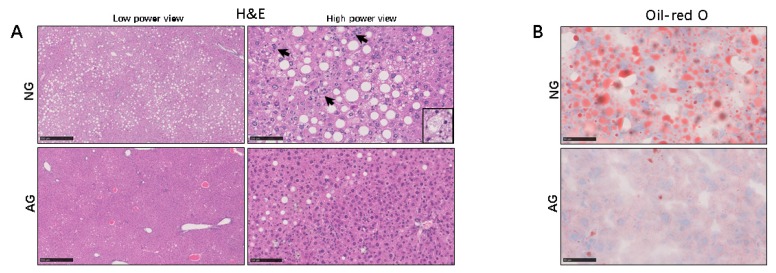
Histological observation of the liver from the two groups of mice. (**A**) Representative photomicrographs of liver (H&E). H&E-stained sections revealed macrovesicular and microvesicular steatosis throughout the entire lobules in the NAFLD livers from NG mice, as well as scattered lobular and perivenular inflammation (arrowhead). (original magnification: ×40 [low]; Bars = 50 μm, ×200 [high]; Bars = 100 μm, *n* = 17). (**B**) Oil Red-O staining revealed a number of lipid droplets accumulated in NG liver, while fewer lipid droplets were observed in the AG liver (original magnification: ×400, Bars = 50 μm, *n* = 17). NG: needling-nonacupoint group, AG: needling-acupoint group, H&E: hematoxylin and eosin, NAFLD: nonalcoholic fatty liver disease.

**Figure 3 metabolites-09-00299-f003:**
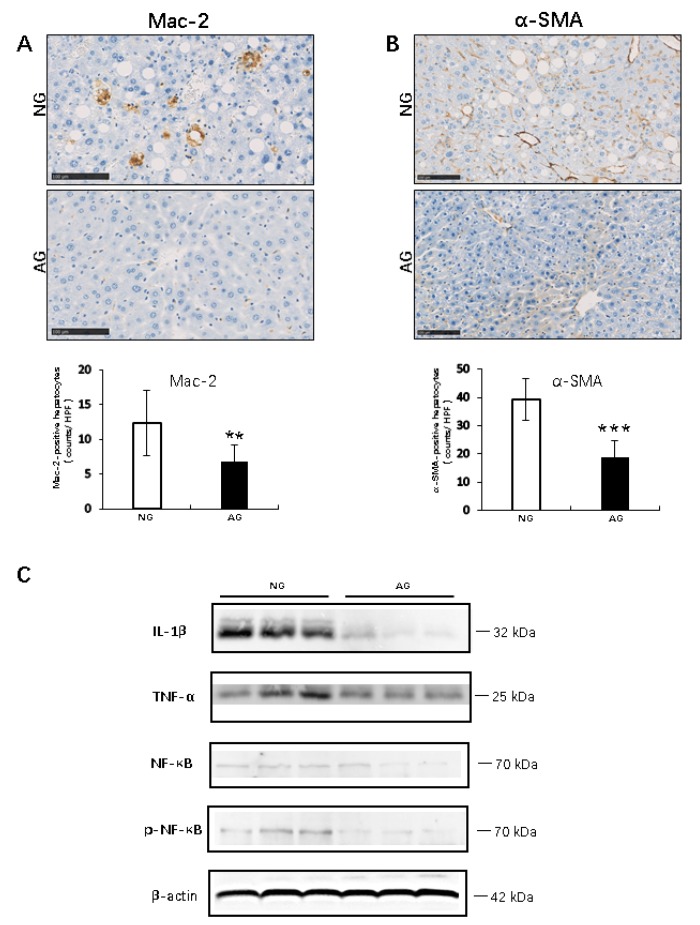
Inflammatory responses in the livers of AG and NG mice. (**A**) Immunocytochemistry (IHC) showed the number of Mac-2-positive infiltrating macrophages in the NAFLD liver. (**B**) α-SMA-positive activated hepatic stellate cells in the NAFLD liver. Values are shown as the means ± SD, Bars = 100 μm * *p* < 0.05, ** *p* < 0.001, *** *p* < 0.0001, *n* = 17. (**C**) The IL-1β, TNFα, NF-κB and p-NF-κB protein expression was determined by Western blotting. Values are normalized β-actin expression (Western blotting) expression and are presented as means ± SD. *n* = 7. NG: needling-nonacupoint group, AG: needling-acupoint group, IHC: immunocytochemistry, α-SMA: α-smooth muscle actin, IL-1β: interleukin 1β, TNFα: tumor necrosis factor-α, NF-κB: nuclear factor-κB, p-NF-κB: phospho-nuclear factor-κB.

**Figure 4 metabolites-09-00299-f004:**
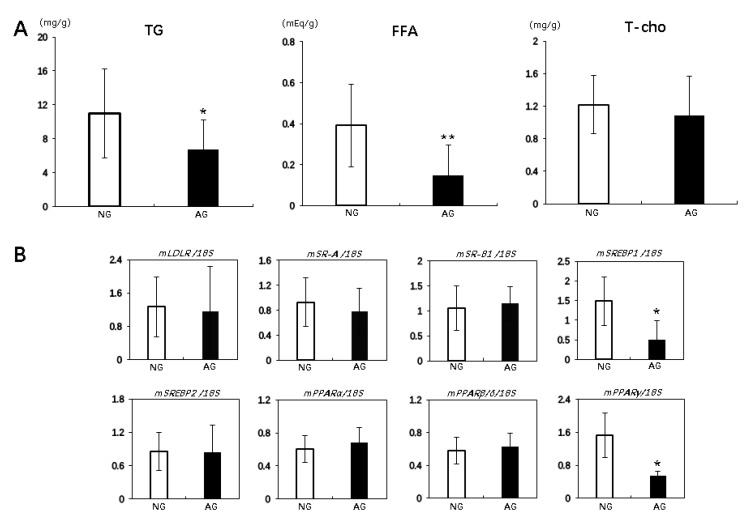
Acupuncture improved the lipid metabolism in the livers of mice with NAFLD. (**A**) A hepatic lipid analysis in NG and AG mice after two weeks of acupuncture. Values are shown as the mean ± SD, * *p* < 0.05, ** *p* < 0.001, *** *p* < 0.0001, *n* = 17. (**B**) Real time PCR revealed that the expression of several proinflammatory signaling factors (SREBP1 and PPARγ) was significantly lower in the livers of AG mice than in those of NG mice. LDLR, SR-A, SR-B1, and SREBP2 showed no significant difference between the groups, nor did PPARα or PPARβ/δ. Values are normalized by the 18S rRNA expression. RT-PCR values are presented as the means ± SD. * *p* < 0.05, ** *p* < 0.001, *** *p* < 0.0001, *n* = 7. NG: needling-nonacupoint group, AG: needling-acupoint group, TG: triglyceride, T-cho: total cholesterol, FFA: free fatty acid, LDLR: low-density lipoprotein receptor, SR-A: scavenger receptor class A, SR-B1: scavenger receptor class B type 1, SREBP: sterol regulatory element-binding protein, PPAR: peroxisome proliferator-activated receptor.

**Figure 5 metabolites-09-00299-f005:**
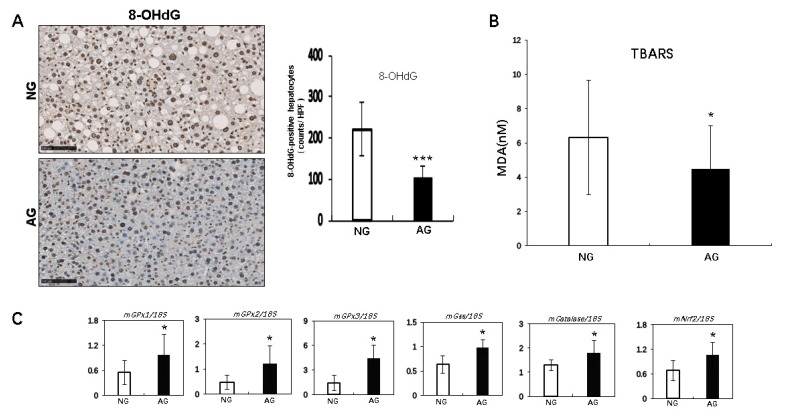
The analysis of hepatic oxidative stress in mice with NAFLD. (**A**) 8-hydroxy-20-deoxyguanosine (8-OHdG) staining revealed significantly fewer accumulated 8-OHdG-positive hepatocytes in the livers of AG mice than in those of NG mice after acupuncture. Values are shown as the mean ± SD, * *p* < 0.05, ** *p* < 0.001, *** *p* < 0.0001, *n* = 17. (**B**) The levels of the oxidative stress marker thiobarbituric acid reactive substances (TBARS) in AG mice were significantly lower than in NG mice after acupuncture. Values are shown as the mean ± SD, * *p* < 0.05, ** *p* < 0.001, *** *p* < 0.0001, *n* = 17. (**C**) Real time PCR showed that the GPx1, GPx2, GPx3, Gss, Catalase, and Nrf2 expression in the liver was significantly higher in AG mice than in NG mice. Values are shown as the means ± SD. * *p* < 0.05, ** *p* < 0.001, *** *p* < 0.0001, *n* = 7. NG: needling-nonacupoint group, AG: needling-acupoint group, GPx: glutathione peroxidase, Gss: glutathionylspermidine synthetase/amidase, Nrf2: nuclear factor erythroid 2-related factor 2.

**Figure 6 metabolites-09-00299-f006:**
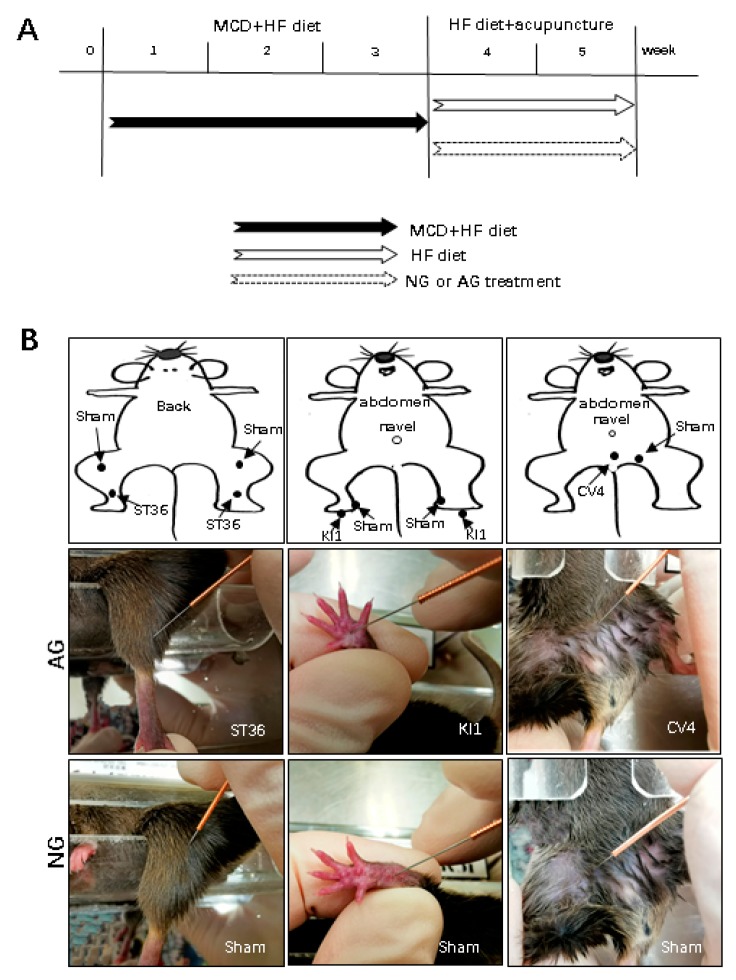
Acupoints and experimental design. (**A**) Schematic diagram of the experimental design. After being fed an MCD + HF diet for three weeks, mice were needled at acupoints or nonacupoints (sham) under a HF diet. (**B**) The acupoints and nonacupoints (sham) are shown in the diagrams (upper pictures). The actual acupoints and nonacupoints (sham) in mice are indicated in the middle and lower pictures. NG: needling-nonacupoint group, AG: needling-acupoint group, ST36: Zusanli, CN4: Guanyuan, KI1: Yongquan, sham: nonacupoints, MCD diet: methionine- and choline-deficient diet, HF diet: High Fat diet.

**Table 1 metabolites-09-00299-t001:** Quantitative scoring of the fat accumulation, inflammation and ballooning in the NAFLD livers of model AG and NG mice.

Steatosis Score			
Score	NG	AG	*P*
0	0	11	<0.001
1	8	6	
2	4	0	
3	5	0	
**Inflammation Score**			
Score	NG	AG	*P*
0	0	5	0.003
1	12	12	
2	4	0	
3	1	0	
**Ballooning Score**			
Score	NG	AG	*P*
0	5	16	<0.001
1	10	1	
2	2	0	
3	0	0	
**NAFLD Score**			
Score	NG	AG	*P*
0–3	7	17	<0.001
4–6	8	0	
7–9	6	0	

Values are shown as the means ± standard deviation, * *p* < 0.05, ** *p* < 0.001, *** *p* < 0.0001, *n* = 17. NG: needling-nonacupoint group, AG: needling-acupoint group, NAFLD: nonalcoholic fatty liver disease.

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
