# Peer review of "Acupuncture on ST36, CV4 and KI1 Suppresses the Progression of Methionine- and Choline-Deficient Diet-Induced Nonalcoholic Fatty Liver Disease in Mice"

_metabolites, 2019, doi:10.3390/metabo9120299_

Round 1

Reviewer 1 Report

In the study entitled “Acupuncture on ST36, CV4 and KI1 suppresses the 2 progression of methionine- and choline-deficient 3 diet-induced nonalcoholic fatty liver disease in mice“, Meng and collaborators aim to investigate the impact of acupuncture in a metabolic syndrome-proned mouse model.

The authors report interesting observations supporting the notion that acupuncture could, at least at the level of the liver, counteract part of the deleterious effect mediated by the aforementionned diet.

Overall, the scientific rational is easy to follow and the data support the claim of the authors.

Yet, I do have the following issues :

1- It is known that a HFD diet without choline and methionine induces severe body weight loss in mice. Therefore, what was the crude body weight of mice in the different groups (HFD/NG vs HFD/AG) after 5 weeks of intervention? Please, provide this piece of information as sup data if the Policy of “Metabolites“ allows it.

2- Did you look at plasma ALT levels ?

3- The authors made the interesting observation that hepatic TG content was less in mice from the AG group compared to that of mice from the NG group. They also assessed for the mRNA levels of PPARs (alpha, beta, gamma) as well as mRNA levels of SREBP-2 and SREBP-1.

However, mRNA levels is not necessarily in straight relation with protein activity.

Therefore, to examine whether HFD/AG (compared to HFD/NG) is associated with increased PPARalpha activity in liver (increased fatty acod oxidation), this reviewer would like to see measurement of mRNA expression of Cyp4A10 and Cyp4A14, both being target genes that are extremely sensitive to the presence and activation of PPARalpha in mouse liver, and can thus serve as markers of PPARalpha activity. SYBR Green based (RT)qPCR should be ok.

4- Following the same idea, did you look at SREBP1 target genes expession levels between HFD/NG vs HFD/AG mice such as ACC1, FAS, SCD1 for instance. This should be done.

5- Expression of PPARgamma targets such as CD36/FAT, LPL, aP2, and uncoupling protein 2 could also be monitored in the livers of both groups of mice (HFD/AG vs HFD/NG), to check whether they mirrored that of PPARgamma.

6- Did you also look at the expression (mRNA levels) of genes involved in hepatic VLDL/TAG secretion (ApoE, ApoB, MTP, ApoCIII) between HFD/NG vs HFD/AG mice? This should/could be done.

7- To strenghten the notion that infiltration (activation) of Kupffer cells is less after acupuncture, this reviewer would like to see the expression levels (mRNA, RTqPCR, liver) of the several critical markers for this specific population such as Clec4f, Vsig4 and to a lesser extent, Cd68 (in addition of the ICH for Mac2, which is already presented).

8- Could you look at the relative expression of fibrosis-related genes in the liver such as Timp1 and Col1alpha1 between HFD/NG vs HFD/AG mice?

9- One reasonnably expect that HFD would cause deterioration of glucose tolerance in HFD/NG mice. Therefore, did you also look at glucose plasma levels to check if HFD/AG mice are protected from hyperglycemia ? What was your unbiased control to be confident that HFD mice indeed trully suffered from a metabolic syndrome after the regimen?

10- Hepatic content of pro-inflammatory cytokines was found to be less in HFD/AG mice. What about any translation of this local decrease in the plasma ? Do you have any data about this ? If so, provide them, please.

11- Table 1 : Please, it is mandatory to provide some explanations in the Mat/Methods section on how (on which bases) the different scores were attributed and by whom (a clinician, pathologist, ? Done by one people only, by two people in blind…. ?).

Minor point :

These results remind us that acupuncture may regulate cell metabolism at the “transcription“ level.

Since you “only“ did RTqPCR experiments, I would change the term “transcription“ by “expression“. Keep in mind that RTqPCR quantifies the number of a special transcript at a particular time ; this number depends on gene transcription but also in parallel, it depends on the half-life of the targeted mRNA. Therefore, it is a balance between gene production (transcription) and destruction by endogenous RNase (half-life).

Reviewer 2 Report

Please see the comment file

Round 2

Reviewer 1 Report

Dear Authors,

Thank you for your efforts to answer to most of my different issues.

I’m almost fully satisfied with your answers, however, I still have three small issues :

1- Point n°5 of your answers:

You claimed that FAS is a PPARgamma target gene,…..this assertion implies that there is a (some) functional PPRE(s) within either the promoter sequence of the Fas gene, and/or elsewehre (5’, 3’ UTR, intronic regions,…).

To the best of my knowledge, it has not been demonstrated so far that Fas is a direct PPARgamma target (in contrast to SREBP), or please, provide me with the right reference from the literature.

What is however known is that some PPAR agonists (Wy14,643) are able to regulate Fas expression in the liver or in mouse/human liver cells (HepG2, Hepa 1.6, FAO,…).

In brief, please, tone-down or better, simply remove the notion that FAS is a PPAR target gene and rather indicate that it is a PPAR regulated gene….(directly or through a molecular intermediate remains to be firmly established according to me…)

2- Cyp4A10 and Cyp4A14 expression was not evaluated by RTqPCR, because you claimed that Pparalpha expression in the liver was unchanged…..perhaps Pparalpha mRNA are unchanged, it does not necessarily means that PPAR activity is unchanged, too.

So, provide me with the RTqPCR data on liver samples for Cyp4A10 and Cyp4A14.

3- How do you interpretate the new RTqPCR data from point 6 and include them into the discussion of the MS ? Higher lipoprotein secretion rate/production by the liver…….means to higher circulating TG ???

Minor points :

Typo lane 161 : «  fat acid syntheas » instead of fatty acid synthase

Typo lane 161 : tearoyl instead of stearoyl

Lane 167 : and the expression of PPARγ targets such as adiponectin receptor 2(AdipoR2).

Change into “as well as the the expression of PPARγ targets such as adiponectin receptor 2(AdipoR2) “.

Lane 170 : hepatic lipids secretion (remove the « s » from lipids)

Lane 170 : Moreover, The expression of genes involved in…. “the“ expression …..

Reviewer 2 Report

This revised manuscript has improved its clarity. Although some responses did not explain the comments. Authors should have incorporated the short but informative responses into the text to make it clear to readers who may have the concerns and interests.
